# Iron Silicides in Fulgurites

**Tian Feng [1], Joshua Abbatiello [1], Arthur Omran [2], Christopher Mehta [1] and Matthew A. Pasek [1,*]**

[1] School of Geoscience, University of South Florida, 4202 E Fowler Ave, Tampa, FL 33620, USA; tianfeng1@usf.edu (T.F.); jabbatiello@usf.edu (J.A.); camehta@mail.usf.edu (C.M.)

[2] Department of Chemistry, University of North Florida, 1 UNF Drive, Jacksonville, FL 32224, USA; n00431947@unf.edu

* Correspondence: mpasek@usf.edu

**Abstract:** Iron silicide minerals (Fe-Si group) are found in terrestrial and solar system samples. These minerals tend to be more common in extraterrestrial rocks such as meteorites, and their existence in terrestrial rocks is limited due to a requirement of extremely reducing conditions to promote their formation. Such extremely reducing conditions can be found in fulgurites, which are glasses formed as cloud-to-ground lightning heats and fuses sand, soil, or rock. The objective of this paper is to review reports of iron silicides in fulgurites, note any similarities between separate fulgurite observations, and to explain the core connection between geological environments wherein these minerals are found. In addition, we also compare iron silicides in fulgurites to those in extraterrestrial samples.

**Keywords:** iron silicides; fulgurites; reducing environment; lightning

## 1. An Overview on Fulgurites and Prebiotic Chemistry

Fulgurites are glassy rocks that are formed when an electric discharge flows through materials such as rock, soil, and sand [1]. The electric discharge in nature is lightning [2,3], but sometimes, a man-made powerline also can be an electric discharge source that produces a fulgurite [4,5]. As a fulgurite forms, a high-energy electric discharge (peak currents as much as 200 kA) quickly (about 100 μs) travels through sand, soil, and clay [6]. This tremendous electric current causes rapid melting of the target material and forms an amorphous, tubular glassy mixture, which traces the path the current traveled through the target.

Fulgurites are categorized as a type of pyrometamorphic natural glass [7–9], where pyrometamorphism is a form of low-pressure, high-temperature metamorphism. Pyrometamorphism is usually surficial, and may be analogous to impact metamorphism, especially given recent discoveries of shock quartz within fulgurites [10,11]. Fulgurites can be partitioned based on differing spark sources and different target minerals [1]: natural fulgurites are those fulgurites formed by lightning hitting materials such as soil, sand, and rock, whereas artificial fulgurites are formed by man-made electrical power structures or sources such as downed powerlines that discharge into natural materials. Anthropogenic fulgurites are formed by natural lightning that strikes artificial substances, such as asphalt or concrete. A fulgurite can be both artificial and anthropogenic if an artificial discharge travels through man-made target material. The major differences between lightning-formed fulgurites and electrical powerline-formed fulgurites are the power and total reaction time. Natural lightning provides a huge, powerful discharge of energy ($10^9$ J per flash) [12], extreme high temperatures (range of $10^4$–$10^5$ K) [13] but limited duration (100 μs) [2,14]. On the contrary, man-made discharge sources have significantly less power than a lightning strike but may stay in contact with a target material for a much longer reaction time (hours). The net result is that it can be difficult to differentiate between artificial and natural sources as both form fulgurites, though glass morphology and composition can provide clues as to what energy source formed them [1].

Fulgurites form when a substantially powerful electrical current strikes and flows through a target mineral. The process requires the creation of an electric arc, which occurs when the electrical voltage surpasses the target material breakdown strength. This electrical arc provides a high-energy, high-temperature, and high-reduction environment that heats the target materials [1]. Furthermore, the high-energy, charged plasma generated by the electric current may heat and expand the tube channel, which may then transfer thermal and kinetic energy into the target material and its surrounding area [15]. The tube size and morphology of the fulgurite depend on the target mineral composition [3] and physical characteristics, as well as the energy and duration of the discharge event [1,15]. The mineralogy of fulgurites reflects their unusual formation conditions and is characterized by highly reduced phases such as iron silicide minerals and iron phosphides (schreibersite), as well as high-temperature minerals such as cristobalite and baddeleyite, and high-pressure minerals such as shocked quartz and cubic $ZrO_2$ [10,11,16–18].

Pasek et al. [3] classified fulgurites by morphology and divided fulgurites into four categories based on glass and crust thickness, and composition. Type I fulgurites consist of sand fulgurites with thin glass walls whereas Type II are clay/soil fulgurites with thick, melt-rich walls. Type III fulgurites are formed in desert soil consisting of $CaCO_3$ cementing sand, with thick, glass-poor walls. Lastly, Type IV fulgurites include rock fulgurites with glass walls surrounded by or occurring on unmelted rock. Exogenic fulgurites are uncommon glassy exhalates that are usually found with Type II and IV fulgurites.

As a natural glass, recent work has suggested that fulgurites may have played a significant role in increasing the amount of reactive phosphorus on early Earth (specifically with relevance to the reduction of phosphorus on early Earth) [19,20]. Copious amounts of reduced state phosphorus compounds, such as phosphides and phosphites, could possibly have been formed by lightning strikes on early Earth [20], thus increasing the availability of the element on Earth's surface. In addition, given the similarity in chemistry between phosphides and silicides (both are formed in highly reducing environments, and both are present in meteoritic samples), constraining the conditions of silicide formation may provide a basis for understanding how phosphides may have also formed and been present on early Earth.

## 2. The Occurrences of Iron Silicides in Fulgurites

During fulgurite-forming lightning strikes, an extremely high-energy and high-temperature environment is generated over the course of a second (the lightning strike timescale), which persists for a few seconds to minutes after the initial strike and cessation of current (the timescale for heat dissipation). These uncommon conditions enable the reduction of oxide minerals into oxygen-poor compounds, such as phosphate minerals transforming into schreibersite $(Fe, Ni)_3P$ and phosphite [19–21]; silicate minerals changing into reduced iron silicides and elemental silicon [2,22–25]; and carbon sources reduced to carbide minerals [26–28].

The iron silicide minerals are like iron phosphides and are a group of rare minerals that formed in reducing environmental conditions. Major natural iron silicide minerals include gupeiite $(Fe_3Si)$ [29], suessite $((Fe, Ni)_3Si)$ [30], hapkeite $(Fe_2Si)$ [31], xifengite $(Fe_5Si_3)$ [28,29], naquite $(FeSi)$ [32], linzhiite $(FeSi_2)$ [33] and luobusaite $(Fe_{0.84}Si_2)$ [34]. Iron silicide minerals are common to a variety of fulgurites and can be found globally. Iron silicides in fulgurites may also be a possible source of the strange occurrences of Fe-Si that are found in unusual locations. We discuss these below, with the caveat that some of these occurrences are within fulgurites with artificial electric discharge sources, or with sources that are ambiguous (where details of the formation event were not recorded, or lost by the observers). Nonetheless, the occurrences of FeSi minerals within fulgurites spans both artificial and natural sources, and indicates the formation of these minerals is not a rare occurrence.

The first occurrence of silicides in fulgurites was reported by Essene and Fisher [2], who discussed the presence of opaque, metallic spherules inside the glassy matrix of

the Winans Lake fulgurite, which was formed by a natural lightning strike, and which showed no evidence of mixing with the much more oxidized matrix glass. This occurrence showed that metal and oxide liquids emerged unmixed at the time of fulgurite formation. Therefore, it was reasoned that the reduced iron silicide minerals, such as FeSi, $FeTiSi_2$, and $Fe_3Si_7$, were formed associated with the fulgurite when these two separate liquids cooled down. Additionally, Essene and Fisher [2] proposed that the reduction of silicates to silicides (and elemental silicon) was coupled to an oxidation of carbon (in the form of graphite, with its precursor likely being a tree root) inside the Winans Lake fulgurite, though the authors suggested oxidation of atmospheric nitrogen may have contributed to the reduction environment.

However, Sheffer [22] argued that this oxidation of carbon or nitrogen provides a fraction of the reducing power needed for the conversion of iron oxide and silica into metal silicides. Sheffer [22] and Roberts et al. [35] investigated several lightning-formed fulgurites and compared the glasses to their organic-poor, starting target materials. They demonstrated that in most fulgurites, iron is reduced compared to the original minerals (averaging 66% reduction of $Fe^{3+}$ to $Fe^{2+}$). Phases more reduced than $Fe^{2+}$ were reported in two fulgurites in the study by Sheffer [22]. In one fulgurite from a sandstone in West Virginia, iron silicide minerals (FeSi, $FeTiSi_2$, and $FeSi_2$) [22,35,36] were identified. Sheffer [22] argued that reductants are not necessary for the formation of these reduced minerals, and instead, the separation of oxygen from oxides by an isentropic, high-temperature heat pulse is sufficient to contribute to the reduced mineralogy of fulgurites. Furthermore, a lightning strike shockwave that hits the target materials, and its prolongation can similarity contribute to the reducing environment of fulgurites. Target mineral grain size and density can directly influence the shock wave propagation and the reduction of fulgurites [35].

Sheffer [22] also provided data on an anthropogenic, artificial fulgurite from Farmington, Connecticut. This fulgurite formed when a 27 kV powerline was struck by lightning in a storm, fell, and discharged into pavement and road gravel. It bore extremely reduced phases such as elemental aluminum and silicon, along with other silicides. Sheffer [22] noted that the silicides in this fulgurite were likely formed as the hydrocarbon tar gluing the asphalt combusted and consumed oxygen.

Cardona et al. [25] investigated a fulgurite formed associated with a downed powerline from El Rosario, Mexico, that bore iron silicides such as $FeSi_2$ and FeSi. The discharge source of this fulgurite was likely both lightning and subsequently the powerline. Cardona et al. [25], building from prior work by Sheffer [24], suggested that that the origin of reduced materials in fulgurites can be cataloged as occurring in four steps: vaporization-reduction-reaction (gas phase)-deposition (solids). The $FeSi_2$ and FeSi formed metallic spherules during the fusion of the soil, and the reducing conditions in El Rosario fulgurite were caused by oxidation of carbon compounds in the form of plant roots.

Pasek et al. [3] determined that the iron silicide from Essene and Fisher [2], Cardona et al. [25], and Sheffer [22] are associated mostly with Type II fulgurites. These iron silicide minerals are primarily more enriched in Si than Fe. However, Pasek et al. [3] also investigated a natural, Type II fulgurite from York County, Pennsylvania, and demonstrated that these reduced iron silicide minerals are primarily rich in iron (Fe > Si). The reduced metal phase in York County fulgurites can be summarized as MN, where M represent metal (Fe, Ni or Ti) and N represent non-metals such as Si or P. The iron silicide materials with formulae such as $Fe_3Si$, $Fe_2Si$, $Fe_5Si_3$, $Fe_7Si_3$, and $Fe_8Si_3$ were identified within the York County fulgurites. Similarly, a fulgurite of ambiguous origins (likely formed when lightning struck the grounding wire of a powerline, which then fell and continued to discharge into soil, which would imply a combination of natural and artificial discharge sources) from Zacatecas, Mexico, also had Fe-rich silicides [37].

Stefano et al. [38] provided a study for a Type II natural fulgurite formed in Houghton Lake, MI, and focused specifically on the silicides present in this fulgurite. Like other Type II fulgurites (per Essene and Fisher [2], Cardona et al. [25], and Sheffer [22]), the iron silicide minerals associate with Houghton Lake fulgurite are Si-rich and include naquite

(FeSi) and linzhiite ($FeSi_2$). However, another Si-rich mineral, luobusaite ($Fe_3Si_7$), was not found in Houghton Lake fulgurite. This is abnormal because, generally, these three Si-rich iron silicide minerals amalgamate at the same highly reduced situation [32–34]. Wang et al. [39] stated that a potential avenue that could be deleterious to the formation of luobusaite is rapid crystallization events such as the creation of synthetic Fe-Si spheres. The Houghton Lake fulgurite is thought to have formed during a similar rapid crystallization event [38]. Thus, the short period over which the fulgurite experienced high temperatures may provide an explanation for the absence of luobosaite in the fulgurite. Furthermore, because of the rapid crystallization (and therefore lack of luobusaite) of the Houghton Lake fulgurite, Stefano et al. [38] suggested that the fulgurite was created by a natural lightning source and not artificial source (i.e., a downed powerline) such as the case of a blue fulgurite found in Michigan [5].

Walter [40] reported a fulgurite also with ambiguous origins found in Oswego, NY, that occurred when lightning struck a power line and followed a nearby guy-wire, and suggested that aluminum iron silicide alloys ($Al_4Si_3Fe$) were formed associated with the exogenic fulgurite. These unusual aluminum and iron silicide alloys were also found in the Farmington, CT, and Zacatecas fulgurites but did not form in spherical aggregates [22,37].

Parnell et al. [41] investigated how silicates reduce into iron silicide (FeSi) during the formation of a natural lightning-formed fulgurite in the Sahara Desert. This fulgurite also bore reduced iron silicides, the first reported occurrence of such material in a Type I fulgurite, which to date remains the only report of iron silicide in sand fulgurites. However, we report here the presence of elemental silicon in a Type I fulgurite from Polk County, Florida (collection details in [15], Raman details in [5,42]), based on a match between the Raman spectra of material within the fulgurite and the known spectrum of Si (Figure 1). In such a fulgurite, it seems likely that iron silicide should also have formed, but due to the paucity of iron in the target material (which is nearly pure $SiO_2$), elemental Si was the only highly reduced phase formed.

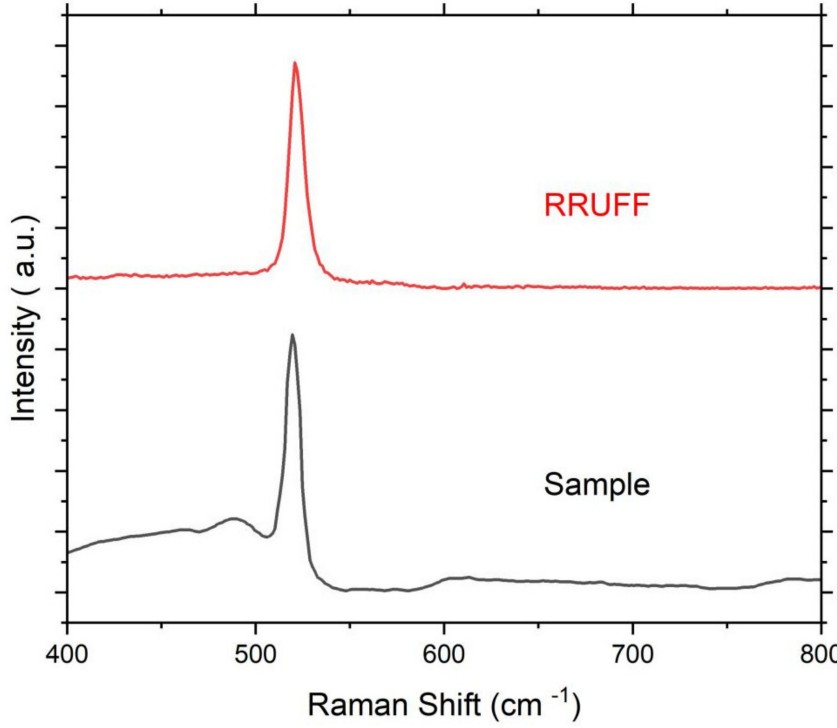

**Figure 1.** Raman spectra of a Polk County, FL, fulgurite with comparison to elemental silicon from the RRUFF database (RRUFF ID: R050145).

In summation, due to the reduced environment that occurs commonly in lightning strikes, iron silicides are frequent accessory minerals in different types of fulgurites. Table 1 summarizes the various reports of fulgurites with reported iron silicide minerals. Generally, the fulgurites that have iron silicide minerals are Type II fulgurites, and the silicides are Si-rich. These silicides occur in a variety of different glasses, from those glasses dominated by $SiO_2$, to those that are enriched in other metal oxides (Figure 2). The two fulgurites with reported occurrences of Fe-Si minerals with more stoichiometric iron (York County and Zacatecas) are both the poorest in Si, indicating that the target composition determines the silicide composition. Note, however, that the number of fulgurites bearing Fe-Si phases is limited to these few occurrences, and the literature on fulgurites is still rather limited. Type III fulgurites have never been reported to contain iron silicides but reducing conditions are still highly plausible in those fulgurites based on the presence of reduced oxidation state P [19]. Exogenic fulgurites are a minor class of fulgurite that are associated with Type II and IV fulgurites, and "erupt" out of a central cylindrical fulgurite.

**Table 1.** Reported fulgurites with iron silicide minerals.

| Fulgurite Name | Fulgurite Catalog (Pasek et al. [3]) | Fe-Si Minerals Reported | Reference |
|---|---|---|---|
| Winans Lake Fulgurite, MI | II | $FeSi$, $Fe_3Si_7$, $FeTiSi_2$ | [2] |
| West Virginia Fulgurite, WV | IV | $FeSi$, $FeSi_2$, $FeTiSi_2$ | [22,35,36] |
| Farmington Fulgurite, CT | II | $FeTiSi_2$, $FeSi_2$, $Fe_2Al_3Si_3$, $Fe_{10}Al_{27}Si_{23}$ | [22] |
| York Fulgurite, PA | II, exo | $Fe_3Si$, $Fe_2Si$, $Fe_5Si_3$, $Fe_7Si_3$, $Fe_8Si_3$ | [3] |
| El Rosario Fulgurite, Mexico | II | $FeSi_2$, $FeSi$, $(Fe, Ti)Si_2$ | [25] |
| Zacatecas Fulgurite, Mexico | II | $Fe_5Si$, $Fe_3Si$, $Fe_2Si$, $FeSi$ | [37] |
| Houghton Lake Fulgurite, MI | II | $FeSi$, $FeSi_2$, $Fe_5Si_3$ | [38] |
| Sahara Fulgurite, Sahara Desert | I | $FeSi$ | [41] |
| Oswego Fulgurite, NY | II, exo | $Al_4Si_3Fe$ | [40] |

The term "exo" is exogenic fulgurite associated with Type II fulgurites.

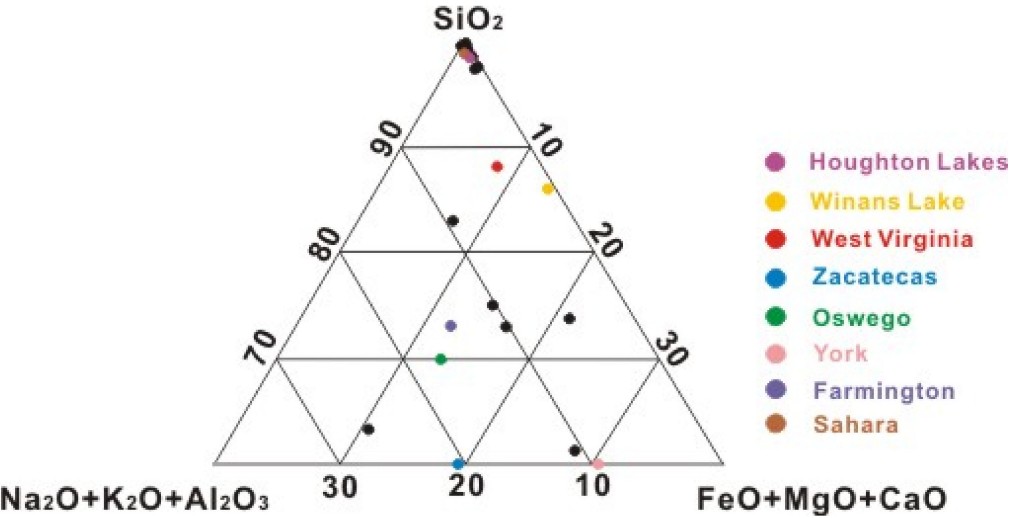

**Figure 2.** Ternary diagram showing the major element (in oxide form, wt. %) of fulgurites glasses of varying composition, based on work in Feng et al. [5]. The fulgurites bearing iron silicides are highlighted in colors other than black [2,3,5,22,35–38,40,41,43,44]. The fulgurite from El Rosario, Mexico, is not shown as major element compositions of the glass were not provided.

### 3. The Formation of Iron Silicide in Fulgurites

The formation of iron silicides is contingent on a reducing environment that provides the necessary conditions for formation of these minerals. Given the strongly oxidizing nature of most of Earth's surface, such conditions are generally rare. The reduction of iron oxide to iron, silica to silicon, $TiO_2$ to Ti, and the FeSi redox mineral buffer are shown as Figure 3.

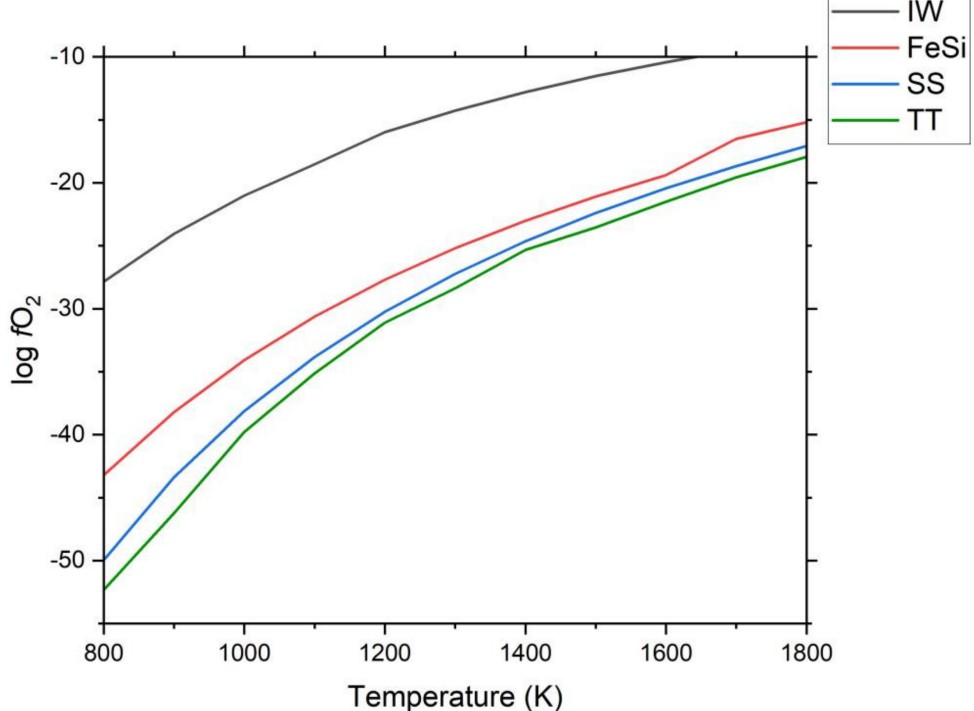

**Figure 3.** Oxygen fugacity mineral redox buffers for materials relevant to iron silicide formation. IW: Iron-wüstite; SS: Scheme 2; TT: Titanium-TiO_2; FeSi: Iron-silicate to iron silicide (Fe + SiO_2 = FeSi + O_2). Data were collected from HSC software (version 9.3.0.9), Oxygen Fugacity Buffer Calculator (Australian National University, https://fo2.rses.anu.edu.au/fo2app/, accessed on 11 October 2021 and Hultgren et al. [45].

Generally, we can group iron silicide minerals in fulgurites into two major groups: silicon-rich and iron-rich. Silicon-rich iron silicide minerals are more commonly found in fulgurites, which in general are much richer in Si vs. Fe [2,19,22,25,38]. Iron-rich iron silicide minerals are more common in extraterrestrial materials, as iron metal is generally more abundant [30,46–49]. However, it is also plausible that fulgurites enriched in iron also contain iron-rich iron silicide [3], and silicon-rich iron silicides can be formed in the solar system [50] (Figure 4).

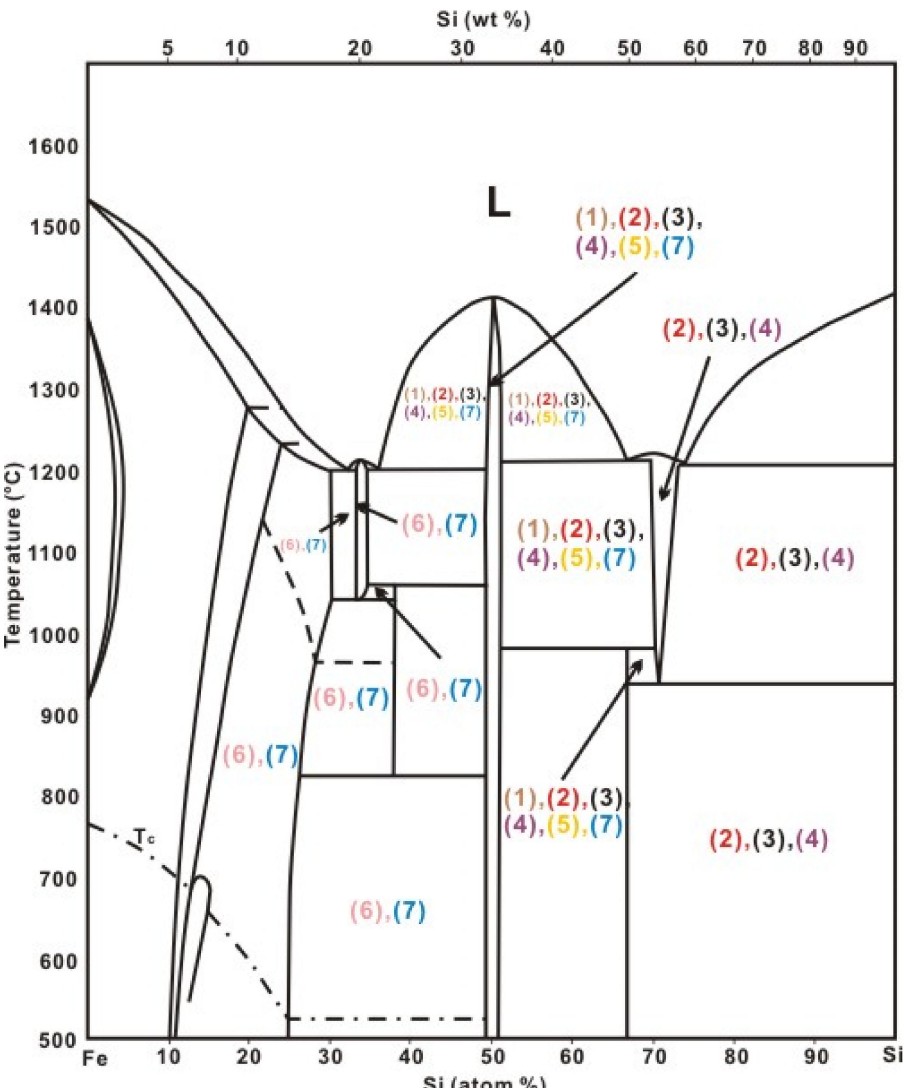

**Figure 4.** Iron-silicon phase diagram. The iron silicides associated with fulgurites are highlighted, which include (1) Winans Lake fulgurite (brown), (2) West Virginia fulgurite (red), (3) EI Rosario fulgurite (black), (4) Houghton Lake fulgurite (purple), (5) Sahara fulgurite (yellow), (6) York fulgurite (pink), and (7) Zacatecas fulgurite (blue). L represents liquid and Tc represents Curie temperature [2,3,22,25,37,38,41,51]. Modified after: Kubaschewski [51].

There have been several formation mechanisms proposed for iron silicide in fulgurites (Table 2). One plausible explanation is reduction with organic carbon. Carbon (such as cellulose from a tree root) can oxidize into carbon monoxide and carbon dioxide at high temperature, donating electrons to the surroundings. Thus, this interaction can reduce silicates and iron oxide into iron silicide [2,25,52]. Furthermore, it is possible that carbon reacts to form carbide with metal, creating the carbides $Cr_3C_2$, $Cr_2C$, SiC, TaC, TiC, and WC [2], though these are not widespread in fulgurites. However, Sheffer [22] argues that this assumption is unlikely for all fulgurites due to the difference between the highly reducing environment required and stoichiometrically low amount of carbon available within most fulgurite target materials (e.g., sand fulgurites bear little to no carbon). However, in environments with heavy vegetation, iron reduction by a "smelting"-like process is likely the prevalent driver for silicide formation [3,22]. In the case of anthropogenic fulgurites, the presence of elemental aluminum (as the main material of some conducting powerlines) may serve as a reducing agent and drive silicide formation.

**Table 2.** Previous research for reduction mechanisms for forming iron silicides in fulgurites.

| Reduction Mechanism | Reference |
| --- | --- |
| Organic Carbon Reduction | [2,3,25,52] |
| Galvanic Reduction | [2,53] |
| Shockwave Chemistry | [10,11,16–18,54] |
| Vapor deposition of Fe and Si | [41] |

Alternatively, the galvanic reduction of target minerals could occur as the electrons flow from clouds to the ground during a lightning strike. This route has been considered with respect to iron silicide formation [2,53]. However, Sheffer [22] challenged this hypothesis, as the electron flows associated with lightning is usually tiny (~30 Coulombs [14]) and it would be difficult to quantitatively promote the reducing conditions necessary to form silicides and reduce iron ($Fe^{2+}$).

Parnell et al. [41] state that high-temperature (>2000 K) liquid vapor deposition and rapid cooling could induce iron silicide formation. Reyes-Salas et al. [37] provide further evidence of vapor deposition in the microscopic morphology of the Zacatecas fulgurite. Pasek and Block [19] propose that at such high temperature, other reduction routes also become plausible, such as calcium phosphate reducing to calcium phosphite, which occurs in Type III fulgurites that typically do not have a significant inner void suggesting that vaporization is not intrinsically required for reduction, but may be important in silicide formation.

A shockwave associated with a lightning strike may also induce a reducing environment and support the formation of iron silicide [10,11,16–18,54]. However, Cardona et al. [25] argues that there is no evidence of shock that has been found in natural fulgurites, which was true at the time. Previous studies show that most fulgurites are characterized by the presence of α-quartz, which is not transformed into stishovite during a lightning strike, which would be expected if the high-temperature excursion was accompanied by high-pressure conditions [55]. Hence the shockwave associated with fulgurite formation may not be up to same level of power as a meteorite impact. Pasek et al. [3] provide the same conclusion that due to the lack of evidence for shock in York fulgurite, this shockwave reduction does not occur in the York fulgurite. However, recent findings suggest shock may be present in some fulgurites [10,11,17].

## 4. Comparison between Iron Silicides in Fulgurites and in Other Rocks

Iron silicide minerals are found throughout the solar system within planets, asteroids, and meteorites [56,57]. Iron silicides are common to the ureilites, which are carbon-rich, achondritic meteorites that have experienced some amount of heating, either from impact or from radioactive decay. Iron silicide minerals were first reported in the North Haig ureilite by Keil et al. [30]. Silicides are also found in at least one Lunar meteorite, which are believed to be formed during impact. Thus, this iron silicide mineral formation route in our solar system requires low oxygen fugacity and ultrahigh-temperature conditions [31,46], which are like the fulgurite-forming conditions [2].

Related to silicides is the dissolution of silicon within iron metal. Silicon is found within metal alloys at up to a mean of ~1.2 atomic % [58] and up to a maximum of 4.6 atomic % in some chondrite meteorites [59]. These Si-rich metals are likely formed through nebular conditions in the solar system, as the equilibrium condensation temperature for this Fe-Si among gaseous and solid phase is 1458 °C at a pressure of $10^{-3}$ atm, solar composition C/O < 0.55 [60]. In contrast to solar system samples, terrestrial silicides are comparatively uncommon. Mantle xenoliths provide some evidence of plausible of iron silicide minerals [61]. The majority Si-rich iron silicide minerals are terrestrial, such as naquite (FeSi), linzhiite ($FeSi_2$) and luobusaite ($Fe_{0.84}Si_2$), that are all found in the mantle podiform chromite deposits in the Luobusha ophiolite, Tibet, China [32–34]. It has been proposed that iron silicides are enriched in ultramafic rocks such as kimberlite [62].

Fulgurites and iron silicides are both products of high-temperature reduction conditions. Hence iron silicides can form within fulgurites [2,3,22]. Given the similarities between fulgurites and impact glasses (e.g., [22]), as well as with nuclear explosions glass such as trinitite, the formation of iron silicides within fulgurites and within meteoritic rocks with evidence of impact seems reasonable.

Worth noting are the differences between cosmochemical and terrestrial silicides. These differences are primarily compositional. Terrestrial iron silicides are usually also enriched in titanium [2,3], and in at least a few cases, may also have aluminum present as part of the structure. However, the aluminum may also have been introduced from the melting of a powerline conductor as conductors are usually made of either Al or Cu. The fulgurites that bear Al-Fe-Si phases (Farmington, CT; Oswego, NY; Zacatecas) are all associated with a powerline, though the specific relationship between powerlines and the fulgurites is not always clear. In contrast, cosmochemical silicides are usually Ti-poor and Ni-rich. Some also bear chromium at up to ~1.5 wt. % [49]. Intriguingly, both terrestrial and cosmochemical silicides bear some amount (0.1–0.5 wt. %) of P as phosphide substituting for Si, which is likely a consequence of phosphorus being similarly abundant in cosmochemical material as it is in soil (~1000 ppm).

Most new, terrestrial iron silicide minerals were found in Luobusha ophiolite, Tibet, China. The Luobusha ophiolite group occurs at a plate suture between the Eurasian Plate and the Indo-Australian Plate [63]. Furthermore, iron silicides have been found throughout the literature as an unusual material that may co-occur with moissanite (SiC) [62,64–67]. Silicides are occasionally attributed to impact, but it is plausible that lightning may be a source of some of these silicides [28]. Some of these iron silicide minerals may even be plausible in the deep mantle and inner core of Earth [68]. These locations indicate that iron silicide minerals may plausibly be present in any environment that has highly reducing conditions [1].

## 5. Conclusions and Future Perspectives

Iron silicide minerals are an unusual and rare mineral group on Earth and are a bit less uncommon in solar system samples. For most solar system samples, the abundance of Si exceeds Fe, but the majority of Si is oxidized in silicates. Iron silicide minerals are hence "rarer" silicon sinks in meteorites. Considering that early Earth atmospheres may have been highly reducing [69], it is possible that iron silicide may have been present on early Earth a bit more frequently than present day and may have served as a prebiotic mineral during the first billon years of Earth's history [42,70,71], perhaps as a driver of reduction. We note here that most fulgurites do not bear iron silicide minerals, and have highlighted the literature occurrences of these materials to the best of our knowledge.

However, due to the rarity of the silicides, little research has focused on determining the connections between iron silicide minerals in various fulgurites. Recent work by Hess et al. [20] highlighted a role for fulgurites in the origin of life. While the role of iron silicides is likely not as prominent in prebiotic chemistry (as neither Fe nor Si is a major component of biomolecules), these materials may still have provided a reactive substrate for promoting redox reactions (e.g., reduction of $CO_2$). Hence, there is merit to further studying iron silicide minerals with respect to the origins of life.

The pathway for iron silicide oxidation is unclear. It is reasonable to assume that iron silicides are ephemeral but may have oxidized into silicate minerals during the first billion years on Earth. Pasek and Lauretta [72] report a water corrosion process with iron phosphide minerals such as schreibersite ($Fe_3P$). Schreibersite can be corroded rapidly by water or organic solvent and transformed into species such as hypophosphite, phosphite, phosphate, and organophosphate compounds [72]. If iron silicide is oxidized by the same pathway as iron phosphide, the products that form after reaction with water or organic corrosion are unclear.

The presence of some unusual Fe-Al-Si minerals in fulgurites may bear a parallel with rocks that have recently been identified to bear quasicrystalline material (Fe-Al-Cu and

Fe-Al-Ni phases such as icosahedrite and decagonite) [73,74]. If the Al is produced through natural processes (and is not a melted conductor wire), then the investigation of Al-Fe phases may yield the first terrestrial examples of quasicrystal minerals that are of wholly natural origin [75].

The formation of iron silicides within fulgurites is testimony to the high-power, highly reducing environment of a lightning strike. By breaking the rules of "normal" mineralogy, iron silicides and similar minerals have caught the eyes of many researchers, both due to their rarity and unusual requirements for formation. The applicability of iron silicides to understanding petrologic processes on Earth and elsewhere has yet to be fully tapped, though future work and investigation into silicide minerals in other fulgurites may help constrain the processes of reduction required for their formation. Additionally, this discussion of iron silicide minerals in fulgurites could help identify fulgurites as being sources of these minerals, instead of necessarily indicating an extraterrestrial origin.

**Author Contributions:** T.F. and M.A.P. conceived and designed the study; T.F. and M.A.P. wrote the draft paper; J.A., A.O. and C.M. reviewed and edited the draft paper; J.A., A.O. and C.M. contributed figures. All authors have read and agreed to the published version of the manuscript.

**Funding:** This research was supported by NASA Emerging Worlds program (Grant 80NSSC18K0598).

**Data Availability Statement:** Data for the Polk County FL fulgurite may be retrieved from the corresponding author's research gate profile, under Research>>Data.

**Acknowledgments:** The authors acknowledge Zachary Atlas, Bolin Yao, and Krystal Cane for providing suggestions to conceive the paper.

**Conflicts of Interest:** The authors declare no conflict of interest.

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
