# Peer review of "Iron Silicides in Fulgurites"

_minerals, doi:10.3390/min11121394_

Round 1
Reviewer 1 Report
The review paper ”Iron Silicides in Fulgurites” by Feng et al. gives an overview on fulgurites and prebiotic chemistry, reports the occurrence of iron silicides in fulgurites, summarizes the formation of iron silicides in fulgurites, and compares iron silicides in fulgurites to those in extraterrestrial samples. Moreover, it gives several future perspectives in studying iron silicide. As far as I know, this might be the first review paper on this topic. Moreover, the draft is well designed and written. The English used is good. I am glad to recommend it to be published in Minerals after the following minor issues are addressed.
The format of refs should be unified.
“(Pasek et al. 2012)” in table 1 should be “[3]”.
“Feng et al. (2019)” in the title of Figure 2 should be “Feng et al. [5]” or just ”[5]”.
Author Response
We thank the reviewer for their review and have modified the references accordingly.
Reviewer 2 Report
It is a very meaningful topic to summarize and compare the types, occurrence state and formation conditions of iron silicide minerals. However, the authors failed to tell the readers under what conditions they are more likely to form, what kinds of mineral assemblages they usually have, and what kinds of early earth or planetary environments they may indicate. Furthermore, besides field samples, the authors rarely mention the research progress of laboratory synthesis to restrict its formation conditions. I cannot recommend publishing it at the current status. There are 5 types of fulgurites reported in literatures (eg. Pasek, 2012), while only 4 are listed in Table 1. Please complete it. Please briefly describe the main differences among them. How to distinguish them? As the authors summarized, most of iron silicide minerals on the Earth may formed under a reduced environment, such as caused by lightning strikes. While there are few descriptions related to the formation conditions of iron silicide minerals outside the Earth. Please make a supplement.Author Response
>>We thank the reviewer for their thorough review, and appreciate their comments.
However, the authors failed to tell the readers under what conditions they are more likely to form, what kinds of mineral assemblages they usually have, and what kinds of early earth or planetary environments they may indicate.
>>The reviewer highlights an important point on our subject that merits careful discussion. Unfortunately, for many of the reported fulgurites which we review, a specific origin (natural vs. artificial source of electric discharge) is not fully clear. In some cases, detailed descriptions of mineralogy and source composition are also lacking. For instance, the Oswego and Zacatecas fulgurites both are associated with powerlines but it is not clear if they were formed by discharge from these powerlines. This is because the finders often bring the samples to lab researchers, who then write up the papers and don’t have the exact details of the formation event (or details are sketchy and inferred). Such ambiguity was acknowledged, for instance, by Stefano et al. (2020), and was looked into for several fulgurites presented by Pasek and Pasek (2018), which sought to distinguish between “natural” and “artificial”. The details that could allow us to differentiate between natural and artificial (size of fulgurites, morphology, presence of conductor wire, formation of cristobalite) are absent in the descriptions we review. We have highlighted that the origin of these fulgurites is “ambiguous”, and better described those that were clearly artificial vs. natural in origin.
We have added some text referencing the above ambiguity to the manuscript.
Furthermore, besides field samples, the authors rarely mention the research progress of laboratory synthesis to restrict its formation conditions. I cannot recommend publishing it at the current status.
>>We acknowledge this difficulty in concisely discussing both topics, but here we have attempted to highlight all known iron silicide occurrences in fulgurites in the literature and give descriptions as to their composition and context of the finding of these fulgurites. In that, we hope the subject can provide details for future researchers looking into FeSi minerals elsewhere, and identify that not all occurrence of FeSi minerals are extraterrestrial.
We have added text to this effect to the final paragraph of the conclusions.
There are 5 types of fulgurites reported in literatures (eg. Pasek, 2012), while only 4 are listed in Table 1.
>> We have added “exo” to the two samples where exogenic fulgurites have been shown to have Fe-Si minerals. These are both associated with type II materials. Type III fulgurites have not been reported with FeSi minerals.
Please complete it. Please briefly describe the main differences among them. How to distinguish them?
>>Distinguishing the types of fulgurites is summarized in ref [3] and expanded in part in ref [1]. Beyond that, the iron silicide minerals are still only found in few definitively natural fulgurites. It may yet be premature to fully state that iron silicides occur under specific conditions or otherwise, until a larger sample suite has been analyzed. We have added text to this effect in the conclusions.
As the authors summarized, most of iron silicide minerals on the Earth may formed under a reduced environment, such as caused by lightning strikes. While there are few descriptions related to the formation conditions of iron silicide minerals outside the Earth. Please make a supplement.
Round 2
Reviewer 2 Report
As the authores clearly point out in the manuscript, the researches on the topic of iron silicides are insufficient both quantitatively and systematically. Therefore, reviews on the reported and latest debelopments are more meaningful. it helps to deeply understand the relevant backfround and the comments on the known and latest progress will help readers to understand the relevant background and expand it to new fields.
All the questions rasised had been well adressed. I would like to recommend accepting this manuscript.